# Safety and Suitability of Infant Formula Manufactured from Extensively Hydrolyzed Whey Protein Compared to Intact Protein: A Combined Analysis of Two Randomized Controlled Studies

**DOI:** 10.3390/nu16020245

**Published:** 2024-01-12

**Authors:** Manja Fleddermann, Anette Knoll, Berthold Koletzko

**Affiliations:** 1HiPP GmbH & Co. Vertrieb KG, Georg-Hipp-Str. 7, 85276 Pfaffenhofen an der Ilm, Germany; manja.fleddermann@hipp.de; 2AK Statistics, Kreppe 2, 85276 Pfaffenhofen an der Ilm, Germany; office@anetteknoll.de; 3Division of Metabolic and Nutritional Medicine, Dr. von Hauner Children’s Hospital, University of Munich Medical Center, Ludwig-Maximilians-Universität München, Lindwurmstr. 4, 80337 Muenchen, Germany

**Keywords:** extensively hydrolyzed whey protein, protein hydrolysate, infant formula, infant nutrition, growth, formula fed, breastfed

## Abstract

Our aim was to assess the nutritional safety and suitability of an infant formula manufactured from extensively hydrolyzed protein in comparison to infant formula manufactured from intact protein (both with low and standard protein content). We performed a combined analysis of raw data from two randomized infant feeding studies. An analysis of covariance (ANCOVA) model was used to determine the non-inferiority of daily weight gain (primary outcome; margin −3 g/day), with the intervention group as a fixed factor and geographic region, sex, and baseline weight as covariates (main model). The data of 346 infants exposed to the formula were included in the analysis. The sample size of the per-protocol analysis with 184 infants was too small to achieve sufficient statistical power. The lower limit of the 97.5% confidence interval (−0.807) of the mean group difference in daily weight gain (i.e., 2.22 g/day) was above the −3 g/day margin (full analysis set). Further anthropometric parameters did not differ between the infant formula groups throughout the study. Growth was comparable to breastfed infants. We conclude that the infant formula manufactured from extensively hydrolyzed protein meets infant requirements for adequate growth and does not raise any safety concerns.

## 1. Introduction

An adequate dietary protein supply is essential for healthy growth and development in infancy [1,2].

Human milk is recognized as the optimal source of nutrients for infants throughout at least the first 6 months of life [3]. However, if breastfeeding is not possible, available, or adequate, infant formulae are the only advisable breast milk substitutes. Infant formulae that are manufactured from cow’s milk protein or other protein sources have been successfully used for many decades.

Next to infant formulae manufactured from intact proteins, infant formulae manufactured from hydrolyzed proteins are available for infants with an increased risk of developing allergies. Hydrolyzed proteins are smaller, easier to digest, and considered to be less allergenic when compared to intact proteins. However, the bioavailability of proteins, amino acids (AAs), and other nutritional components may be different between infant formulae manufactured from hydrolyzed protein (HP) formulae and intact protein (IP) formulae [4]. Based on the degree of hydrolysis and the proportion of small peptides, hydrolysates are classified as partially or extensively hydrolyzed.

Several studies evaluating HP formulae with standard or low protein content and using extensively or partially hydrolyzed whey or casein proteins indicate that HP formulae are safe in terms of growth when compared to human milk, intact cow’s milk protein formulae, or growth standards [4,5,6,7,8,9,10,11,12,13,14].

Although the use of protein hydrolysates has been permitted for many years and the use of protein hydrolysates in the manufacturing of infant formulae is widespread in the market, the level of hydrolysis, the protein source, and other components may affect the safety and tolerance of different HP formulae; thus, the extrapolation of results from one HP formula to another is not accepted by regulators [15].

The growth effects and safety of infant formulae marketed by HiPP (Pfaffenhofen, Germany) compared to human milk have been evaluated in two studies: the HA study [16] evaluated different HP formulae with standard and low protein content, whereas the BeMIM study [17] compared IP formulae with standard and low protein content. In both studies, the standard and low-protein formulae manufactured from either protein hydrolysate or intact protein were considered safe and suitable for infants up to the age of 4 months.

To further explore the nutritional safety and suitability of these HP formulae comparatively, we compiled measures of growth between the HP formula and IP formula by performing a combined analysis of the HA and BeMIM studies.

## 2. Materials and Methods

### 2.1. Analysis Approach and Individual Studies

The HA and BeMIM studies were randomized, controlled, and double-blinded and investigated the non-inferiority of low-protein infant formulae to conventionally used infant formulae with standard protein content. While the HA study [16] evaluated infant formulae manufactured from hydrolyzed proteins, the BeMIM study [17] focused on IP formulae. Both studies used a non-randomized breastfed group as a reference. The intervention period of study formula feeding for each participant lasted from birth (starting, at the latest, from 1 month of life) until 4 months of life and included monthly study visits.

The combined data analysis was performed based on raw data from the two studies. The inclusion criteria differed slightly between studies and were aligned post-hoc to allow for comparable data. Healthy term newborns ≤ 28 days of life with a gestational age of ≥37 weeks and a birth weight between 2500 and 4500 g were included in the combined analysis.

### 2.2. Diet

The HP and IP formulae compared had similar protein content at the respective protein level (i.e., standard versus vs. low protein). HP formulae were whey-based, extensively hydrolyzed, and included 1.9 g protein/100 kcal (LP) or 2.3 g protein/100 kcal with or without synbiotics. For the synbiotics, *Limosilactobacillus fermentum* CECT5716 was used as a probiotic and galacto-oligosaccharides (GOS) as prebiotics. IP formulae were based on a whey:casein ratio of 60:40 and protein content of either 1.9 g/100 kcal (LP) or 2.2 g/100 kcal.

Both the HP and IP formulae were supplemented with individual AAs to meet regulatory requirements. Except for the IP formula containing 2.2 g/100 kcal, all formulae contained arachidonic and docosahexaenoic acid in a ratio of 1:1. For more information on the infant formulae compositions, see Appendix A.

Five intervention groups were investigated. From the HA study, eHF—infant formula manufactured from extensively hydrolyzed whey protein and LPeHF + Syn—low-protein infant formula manufactured from extensively hydrolyzed whey protein with synbiotics; from the BeMIM study, iPF—infant formula manufactured from intact protein and LPiPF—low-protein infant formula manufactured from intact protein; and, as a reference in both studies, BF—breastfeeding.

Infant formulae were administered orally ad libitum. Infant formula intake, as well as the number of breastfeeding meals and intake of other food or drinks, e.g., energy-containing liquids (sweetened tea or juice) or non-energy-containing liquids (tea and water), were documented using 3-day protocols.

### 2.3. Primary and Secondary Outcome Assessments

The primary outcome was the average daily weight gain in grams per day (g/day) between 1 and 4 months of life, to estimate adequate growth. Secondary outcomes included measurements of body length and head circumference, including respective z-scores as further growth indices, and nutrient intake, adverse events, stool characteristics, and biochemical markers (blood urea, albumin, AAs).

The aim of the combined analysis was to compare HP with the IP formula using formulae with standard protein content (i.e., eHF vs. iPF) and formulae with low protein content (i.e., LPeHF + Syn vs. LPiPF).

### 2.4. Statistics and Power Estimation

Retrospective sample size estimation scenarios were used to determine if sufficient infants were available in a confirmatory setting. Between 43 and 94 participants per group for LPeHF + Syn vs. LPiPF and eHF vs. iPF, respectively, were required to demonstrate non-inferiority. The power simulations were based on the means and standard deviations (SD) of the groups used in the HA and BeMIM studies, using both a per-protocol set (PPS) and full analysis set (FAS), a non-inferiority margin of −3.0 g/day at a one-sided significance level of 2.5%, and a power of 80% (Appendix A). 

The FAS comprised all enrolled participants who participated at least in the month 1 visit and had received study formula. In PPS, only data from participants complying with the predefined conditions, such as completion of the intervention period up to 4 months of life, no intake of other infant formulae besides the study intervention, and breastfeeding at a maximum of once daily, were included. According to local practice, in the study countries, some infants receive liquids, like tea or water, in addition to breastmilk or infant formula, during the first 4 months of life, which was limited to a maximum amount of 50mL per day to still be included in the PPS. The main conclusions on the primary outcome measure were based on PPS; FAS served as a sensitivity analysis.

For the analysis of the daily weight gain, an analysis of covariance (ANCOVA) model was used to show non-inferiority, with the intervention group as a fixed factor and region, sex, and baseline weight as covariates (main model). An ANCOVA with only the intervention group as a fixed factor was used as a sensitivity analysis. Furthermore, additional covariates (maternal age at infant’s birth, maternal body mass index (BMI), gestational age, smoking status of mother before and during pregnancy, weight at birth, maternal education, socioeconomic status) were included in the model. However, these additional covariates did not reveal any plausible and consistent relations with daily weight gain over all populations and group comparisons (LPeHF + Syn vs. LPiPF; eHF vs. iPF) in the ANCOVA model; thus, the results are not described.

The lower limit of the 97.5% confidence interval (CI) of the difference between formula groups was compared to the non-inferiority margin of −3.0 g/day. A hierarchical test design was assumed with ordered hypotheses (Step 1: eHF vs. iPF, Step 2: LPeHF + Syn vs. LPiPF), which was the only analysis that took multiple testing into account.

The European Food Safety Authority (EFSA) Guidance [18] stipulates an alternative way to analyze adequate growth, i.e., the equivalence in growth between intervention groups. Thus, this was also tested using an ANCOVA model, with the region and baseline value as covariates and weight-for-age z-scores at the age of 4 months. Equivalence was concluded if the calculated two-sided 90% CI of the estimated mean difference in the weight-for-age z-score was within the predefined margin of ± 0.5 SD, a bandwidth considered to be indicative for adequate growth.

Secondary outcome analyses were carried out in the PPS and FAS. Daily length and head circumference gains were compared between intervention groups using ANCOVA models similar to the model used for the primary outcome, with respective baseline characteristics included as covariates, but with a focus on superiority.

z-scores were calculated based on the World Health Organization (WHO) growth standards for breastfed children [19]. Comparisons between intervention groups for z-scores were done using a mixed model of repeated measurements (MMRM), with the region, visit, intervention group, and intervention group-by-visit interaction as fixed factors and participant as a random factor, as well as additional covariates (Appendix A).

Nutritional parameters (intake of study infant formula, other infant formula, energy-containing liquids, complementary feeding, and additional breastfeeding) as well as biochemical markers were evaluated using the van Elteren test adjusted for region. A Cochran–Mantel–Haenszel test, adjusted for region, was used to analyze gastrointestinal tolerance. The analysis of adverse events evaluated the number and frequency of intervention-emergent events by system organ class and preferred terms (according to the Medical Dictionary for Regulatory Activities (MedDRA) coding version 23.1). Amino acids and laboratory parameters were descriptively assessed and two-sided superiority tests on a significance level of 5% were applied.

Linear regression models were used to examine the dependencies between average liquid intake per day and weight gain, or infant formula intake between month 1 and month 4.

Statistical analyses were performed with the WPS Workbench version 4.3 (^©^ Copyright World Programming Limited 2002–2022) using the SAS language code. 

### 2.5. Study Population 

The HA study was conducted between 2010 and 2013 as a multicenter study in Germany, Austria, and Serbia, and the BeMIM study between 2010 and 2011 as a single-center study in Serbia. Overall data from 1008 participants were available, of which 589 participants were included in the combined analysis (385 randomized formula-fed participants and 204 breastfed participants). The main reasons for the exclusion of 419 infants from the analysis were receiving infant formulae from a different protein source (see study design of Ahrens et al. (2018) [16], a missing visit at month 1, or screening failures. 

To evaluate HP vs. IP formulae, the data of 385 randomized formula-fed participants were considered, of which 353 participants took part in the 1-month visit, had documented data, and received at least one bottle of study formula, and 307 completed the 4-month follow-up visit (Figure 1). The FAS comprised 346 participants (86 eHF, 89 iPF, 83 LPeHF + Syn, 88 LPiPF). For PPS, data were limited to 184 participants (39 eHF, 42 iPF, 54 LPeHF + Syn, 49 LPiPF). The predominant reasons for exclusion from PPS in the formula groups were the violation of the feeding regimen and an age outside the visit or enrolment window; not being allowed concomitant medication; early withdrawal; and the violation of the eligibility criteria. Data from 204 breastfed participants (203 FAS, 115 PPS) served as an external reference.

Baseline characteristics including sex, first-born status, maternal and paternal age and BMI, maternal smoking, age at randomization, mode of delivery, and anthropometry at birth were similar between the eHF and iPF groups and between the LPeHF + Syn and LPiPF groups for FAS (Appendix A) and PPS, except for education, where mothers in the iPF group had higher education levels compared to those in the eHF group (FAS). Due to the different study designs (multicenter vs. single center), all participants in the iPF and LPiPF groups were form Serbia, while over 66% of participants in the eHF and LPeHF + Syn groups were from Serbia and 33% from Germany and Austria.

## 3. Results

### 3.1. Weight Gain and Growth 

The mean difference in daily weight gain in participants receiving eHF compared to participants fed iPF was 0.73 g/day (CI [−3.029, inf.]) for PPS, with the lower limit of the 97.5% CI narrowly missing the predefined non-inferiority margin of -3 g/day (Figure 2, Appendix A).

In a sensitivity analysis using FAS, the lower limit of the 97.5% CI (−0.807) of the mean group difference in daily weight gain (i.e., 2.22 g/day) was well above the −3 g/day margin (Figure 2, Appendix A). Similar results were seen in a further sensitivity analysis using an ANCOVA model without adjustments, yielding lower limits of the 97.5% CI above the −3 g/day margin (PPS: −2.059; FAS: −1.512, Figure 2). 

The difference in daily weight gain between the LPeHF + Syn and LPiPF groups was 1.39 g/day (CI [−1.321, inf.[) in PPS, and 0.28 g/day (CI [−2.344, inf.[) in FAS (Figure 2, Appendix A). Similar results were obtained using an ANCOVA without adjustments (lower limit of the 97.5% CI, PPS: −0.606; FAS: −2.068, Figure 2). According to the hierarchical test scheme, the procedure stopped at the inferiority testing of eHF vs. iPF and no further inferential conclusions could be made when testing the LPeHF + Syn vs. LPiPF groups.

The length gain from months 1 to 4 did not differ between the HP and IP formula groups in PPS. Head growth was greater with LPeHF than LPiPF (*p* = 0.0192), but similar between eHF and iPF (PPS). The FAS analyses showed comparable results except for a significant difference in length gain between the eHF and iPF groups (*p* = 0.0325) (Appendix A). In BF participants, the gains in weight, length, and head circumference were smaller or similar to those observed in the formula groups (Appendix A). 

Anthropometric measurements, expressed as z-scores (Figure 3 for PPS, Appendix A for FAS), were within -1 to 1 from months 1 to 4 of life, confirming age-appropriate development in all formula groups. No differences were observed between HP and IP formulae at any time between 1 and 4 months of life. For both PPS and FAS, an MMRM analysis confirmed that there were no differences between intervention groups (Appendix A).

An ANCOVA at the age of 4 months confirmed equivalent growth with HP and IP formulae, i.e., the two-sided 90% CI of the mean difference in the weight-for-age z-score was contained within the pre-defined equivalence margin of ± 0.5 SD, in both FAS (eHF vs. iPF: [−0.140; 0.413]; LPeHF + Syn vs. LPiPF: [−0.346, 0.132]) and PPS (eHF vs. iPF: [−0.357, 0.327]; LPeHF + Syn vs. LPiPF: [−0.207, 0.280]).

### 3.2. Nutrient Intake 

No differences in energy intake of the study infant formula between eHF and iPF were found throughout the observation period (PPS and FAS). The average energy intake in the LPeHF + Syn and LPiPF formula groups was comparable at months 2 and 3, but significantly higher in the LPeHF + Syn group at month 1 (FAS) and month 4 (FAS and PPS, Appendix A).

The number of infants with documented additional breastfeeding was lower in infants fed the HP formula (Appendix A). The same applied for energy-containing liquid intake (Appendix A).

The number of infants in both the HP and IP groups who consumed other formulae and/or complementary food was too low (at a maximum of six infants) for a meaningful comparison (for FAS and PPS). Complementary feeding, generally, did not start before 4 months of life.

### 3.3. Impact of Liquid Intake on Growth and Formula Intake 

Energy-containing liquids were consumed by 42.2% of infants in the formula-fed groups (FAS) and 28.6% of breastfed participants. No measurable effects of energy-containing liquid intake on participant weight gain were observed, as evidenced by a broad scatterplot and a Spearman’s correlation coefficient near zero (Figure 4A). There was also no correlation between the intake of liquid and study formula for energy in kcal/days (Appendix A), as well as for the amount of intake in mL/day (Appendix A). In line with this, the weight-for-age z-scores did not correlate with energy-containing liquid intake at months 1, 2, 3, or 4 (Figure 4B). In addition, the influence of mean energy-containing liquid intake on the amount of infant formula intake from month 1 to month 4 could not be confirmed. Spearman’s correlation coefficients were around zero (Appendix A). Most infants consumed less than 50 mL/day of energy-containing liquids. As for the overall population, there was no obvious influence of energy-containing liquid intake on weight gain in infants consuming less than 50 mL/day of energy-containing liquids (Figure 4).

### 3.4. Suitability

#### 3.4.1. Adverse Events

The percentage of infants affected by adverse events was comparable in each intervention group and the BF group and no formula-related risks were observed (Appendix A). The incidence of serious adverse events was between 2.3% (iPF) and 6.8% (LPiPF), but none of the serious adverse events was related to infant formula intake. Adverse events associated with infections were more frequent in IP than in HP formula-fed participants, while pyrexia appeared to be more common in the HP formulae group. Overweight, which was documented in more detail in the HP study, was observed in HP formula-fed participants (Appendix A). An MMRM analysis, however, did not indicate any differences in growth (weight-for-age and BMI-for-age z-scores) between the HP and IP groups from 1 to 4 months of life (Appendix A).

#### 3.4.2. Stool Characteristics

Stool characteristics were documented over a period of three days prior to each visit. For FAS, significantly more HP-fed infants showed a lower stool frequency than IP-fed infants at 1, 3, and 4 months (*p* ≤ 0.001, eHF vs. iPF) and at 2, 3, and 4 months (*p* < 0.001, LPeHF + Syn vs. LPiPF, Appendix A). 

“Green” colored stools were reported more frequently in the HP formula groups (with increasing frequencies over the observation period; eHF: from 15.4% at month 1 to 34.2% at month 4 of life; LPeHF + Syn: from 27.8% at month 1 to 60.0% at month 4; FAS) compared to IP formula groups (iPF: 2.0% to 7.8% with the highest value at month 3, LPiPF: 4.9% to 7.5%, highest value at month 2; FAS). Statistically significant different stool color patterns were reached between the low-protein groups (LPeHF + Syn vs. LPiPF) at all timepoints in FAS (Appendix A). The stool color patterns between the standard protein groups (eHF vs. iPF) were only statistically different at month 1 in FAS. In the BF group, green stools were reported at a similar frequency as in the IP groups (3.6% to 9.9% (at month 2) of infants). In the BF and IP groups, most infants (>90%) reported stools ranging from yellow and brown to mustard-like during the entire observation period, while, in the HP groups, infants presenting these stool colors decreased over time (eHF: from 84.6% at month 1 to 64.7% at month 4 of life; LPeHF + Syn: from 71.8% at month 1 to 38.8% at month 4; FAS). Black/grey stools were hardly seen in any group (generally <1.5%, except for the eHF group at month 3 with 3.1%, FAS). 

No significant differences between groups were observed for stool consistency in FAS (except for LPeHF + Syn vs. LPiPF at month 3 in FAS, *p* = 0.019). The stools of HP-fed participants were more frequently described as “watery” (eHF: 9.4% to 14.6% and LPeHF + Syn: 5.2% to 10.6% across observation period; FAS) compared to the stools of IP formula-fed participants (iPF 0.4% to 3.9%, LPiPF 0.8% to 3.5%; FAS), but most infants (85% to 95%, FAS) had a stool consistency ranging from soft, formed, sausage, and soft sausage to mushy stools (Appendix A). “Watery” refers only to the stool consistency and does not include diarrhea. The proportion of infants with “watery” stools tended to increase from month 1 to month 4 in all formula groups. Infants with hard stools were rare in the eHP and iPF groups (below 1%, except for the iPF group at month 1, 3.9%; FAS) but slightly more common in the LPiPF group (2.3% to 7.4%, FAS). Compared to the formula groups, BF infants reported more frequently watery stools (21.2% to 31.4%); stool consistencies ranging from soft, formed, sausage, and soft sausage to mushy were presented by 68.6% to 78.6% of BF infants and hard stools were rarely seen (≤0.2%).

#### 3.4.3. Biochemical Markers

The values for plasma albumin and blood urea nitrogen (BUN) were within normal ranges (3.0–5.2 g/dL for albumin, 2.0–7.2 mmol/L for blood urea nitrogen [20]) for most participants in all intervention groups at month 4 in PPS and FAS (Table 1). While the BUN values did not differ significantly between the HP and IP formula groups, HP formula-fed participants had significantly higher plasma albumin values compared to respective IP participants (van Elteren test adjusted for region) in both PPS and FAS. BUN tended to be higher in the formula groups than in the BF group. While 5% (eHF) to 21% (LPeHF + Syn) of formula-fed infants had BUN concentrations below the reference range reported by Oster [20], more, i.e., 52%, of breast-fed infants had BUN concentrations below the reference range.

Amino acid plasma levels were evaluated at month 4. Although the AA profile appeared to be similar in all groups (Appendix A), the plasma concentrations of most AAs were significantly lower in the IP compared to the respective HP formula-fed participants. Exceptions were glutamic acid, ornithine, and phenylalanine, which were comparable between eHF and iPF, and aspargic acid, proline, and valine, which were comparable between LPeHF + Syn and LPiPF (FAS, Appendix A). 

## 4. Discussion

We conducted a combined analysis of two randomized studies in healthy term infants to compare the effects of an HP formula to an IP formula on growth parameters during the first 4 months of life. The results indicated non-inferior weight gain between infants consuming formulae manufactured from extensively hydrolyzed whey protein at standard or low protein levels and infants receiving formulae manufactured from intact cow’s milk with comparable protein content. The non-inferiority margin (−3 g/day for the lower limit of the 97.5% CI) for daily weight gain was reached, when comparing eHF vs. iPF (FAS) and LPeHF + Syn vs. LPiPF (PPS, FAS). For the eHF vs. iPF (PPS) comparison, the non-inferiority margin was narrowly missed (lower 95% CI: −3.029), presumably due to the low number of infants and inadequate sample size in PPS (39 eHF and 42 iPF vs. 94 per group required). Secondary endpoints of growth indices, i.e., weight, length, and head circumference, were generally similar between HP and IP formulae at both protein levels, except for greater monthly head circumference growth (low-protein infant formula, PPS and FAS) and length gains (standard protein level, FAS) in the HP compared to IP formula. However, compared to the WHO growth standards, the mean z-score values of all intervention groups were within ±1 SD during the intervention (Figure 3, in both PPS and FAS and at both low and standard protein levels), indicating comparable and adequate growth with all interventions. In addition, no significant differences between intervention groups were observed for all other z-score values assessed in this study, suggesting that the few observed differences were not clinically relevant.

The findings of this study are consistent with observations reported in other publications. Karaglani et al. (2020) [9], Picaud et al. (2020) [4], and Otten et al. (2023) [21] demonstrated in three randomized controlled studies comparable growth between infants fed a partially or extensively hydrolyzed whey-based formula and infants fed an intact cow’s milk protein formula during the first 4 to 5 months of life. Similar results with no difference in growth characteristics between HP and IP formulae were observed by Wu et al. (2017) [22] in healthy term infants from enrolment to 7 and 13 weeks of life. A pooled analysis of seven clinical studies compared intact cow’s milk infant formulae to a partially hydrolyzed whey infant formula from a single manufacturer on growth at 2 weeks and 1, 2, 3, and 4 months of life [23]. There were no differences in weight gain between infant formula groups. In contrast, in a rather small study (56 infants) performed by Mennella at al. (2011) [10], infants fed an HP formula had significantly lower weight-for-length z-scores compared to IP formula-fed infants across ages 2.5 to 7.5 months. As discussed by others [9], this difference may be due to the lower food consumption observed in this study in the HP formula group. HP formulae contain peptides that can display a bitter taste [24] and might also lead to more rapid satiation [25,26].

Despite recent improvements in infant formulae composition, formulae still contain slightly higher levels of protein than human milk, associated with increased rates of weight gain [27]. Consistent with other findings [28,29,30], we observed an increase in weight-for-age or BMI-for-age z-scores during the first 4 months of life in all formula intervention groups, while no increase was seen in BF infants. 

The LPeHF + Syn that was compared with the LPiPF formula in our analysis contained additional synbiotics (combination of L. fermentum and GOS), making a direct comparison difficult. Results from other studies and meta-analyses do not indicate an impact of synbiotic-supplemented formulae on growth [31,32].

Generally, there were no differences in energy intake from the study infant formula between HP and IP formulae, except for significantly lower total energy intake in the IP formula group at the low protein level at months 1 and 4. However, this difference may be due to the higher breastfeeding rates in the IP formula groups, which is an additional energy source for the infant. Other studies did also not see a consistent difference in infant formula intake between HP and IP formulae. Karaglani et al. (2020) [9] observed higher weekly infant formula consumption (~+10.5%) in IP compared to HP formulae-fed infants but this difference disappeared when daily infant formula intake was corrected for body weight. Czerkies et al. (2018) [23] reported a more pronounced increase in infant formula intake over time in the HP than in the IP formula group, a difference that, however, was only evident among girls.

The number of infants consuming additional formulae and/or complementary food was low in all intervention groups, with no apparent differences. The consumption of energy-containing liquids was higher in the IP than HP formulae groups, but this, however, had no impact on growth or study formula intake. One factor that may have contributed to the differences in energy-containing liquid intake may be their more common use in Eastern European countries, as previously reported by Schiess et al. (2010) for Poland [33]. In our analysis, data for the IP groups were derived from the BeMIM study and thus solely from Serbia, whereas the HP formulae study was performed in Serbia, Germany, and Austria. Infants fed an IP formula (low or standard protein) received also more additional breastfeeding than HP-fed infants. Energy intake from breastfeeding could, however, not be quantified because the amount of breastmilk per meal and its composition were not recorded and analyzed. 

In general, any additional intake of energy-containing liquids is considered a protocol deviation in infant growth studies. We therefore conducted a correlation analysis on the impact of energy-containing liquid intake on infant formula intake and weight-for-age z-scores. In contrast to Schiess et al. (2010) [33], who reported that infants receiving energy-containing liquids had an approximately 30 kcal lower infant formula intake based on data from a multicenter European study, the intake of liquids up to 50 mL/day did not correlate with the amount of or energy intake from the infant formula in our analysis. In the Schiess study, an inverse relationship between liquid/tea intake and energy intake from infant formulae was observed. Reasons for these inconsistent results are unclear, but they may arise in part from the different analysis approaches used: while Schiess et al. (2010) compared groups with and without energy-containing liquid intake with a Wilcoxon rank-sum test, we used a linear regression model/correlation measures to investigate the connection between growth, formula intake, and energy-containing liquid intake for all infants. The average energy intake in kcal/day from liquids was comparable between our analysis and the Schiess study. In the latter, the impact of energy-containing liquid consumption on growth was not evaluated, but, in our analysis, liquid intake (up to 50 mL/day) had no visible positive or negative effect on infant weight gain during the observation period. No conclusion can be drawn for liquid intake of more than 50 mL/day, as the number of participants who consumed more than 50 mL liquid/day was very low. The proportions of infant formula-fed infants consuming energy-containing liquids were comparable in our analyses and the Schiess study (about 42%), but differed for BF infants (29% in our analysis vs. 10% in the Schiess study). 

Concerning adverse events, around a quarter of infants experienced adverse events, with no differences between intervention groups. No serious, formula-related adverse events occurred. A higher incidence of adverse events related to the study formula was reported for the HP formula groups versus their IP counterparts, but given the transient nature and mild severity, this was not considered a safety concern.

There were also no consistent differences in tolerance parameters between the HP and IP formula groups, in line with findings from other studies [5]. The reduced stool frequency observed for the HP formula was unexpected and contrasts with previous findings that report an equal or increased stool frequency [5,22]. HP formulae are supposed to shorten the gastrointestinal transit time [34,35], generally associated with an increased stool frequency. The trend towards more watery stools in the HP formula groups than in their IP formula counterparts aligns with a shorter transit time and is compatible with other findings [4,23]. The higher prevalence of “green” stools in the HP formula in our analysis is commonly seen in infants consuming HP formulae [36], which might be explained by the hydrolyzed proteins, which are absorbed and metabolized differently from intact proteins [37].

The plasma AA levels observed in our analysis generally reflected the composition of the different infant formulae tested, with lower AA levels in the IP than in the HP formula. The IP compared to the HP formula groups reached plasma concentrations closer to the BF reference, reflecting a more balanced AA intake typical for IP formulae due to a whey:casein ratio close to human milk. Glutamine and cysteine plasma concentrations were below the BF reference in both IP groups. Following the EU guidance on infant formula compositions, the amount of cysteine present in infant formulae can be summed up with methionine (if the ratio of the two amino acids is less than 2); thus, cysteine values must always be evaluated in light of methionine intake. While the plasma cysteine concentrations were below those of the BF reference group in the IP groups, the median methionine concentrations were above those of BF infants. The methionine concentrations in formulae were adequate and thus the sum of both AAs was in line with the EU guidance. Glutamine and glutamic acid and serine levels in infant formulae are not regulated by EU law. However, human milk contains significant amounts of glutamate, which has been suggested to be important for intestinal development in infants [38,39,40,41,42]. 

BUN values did not differ between the HP and IP formula-fed groups. Our results showed that a much larger proportion of breastfed infants had BUN values below the reference values than formula-fed infants. It appears that the reference ranges should be redefined based on the levels observed in healthy, growing breastfed infants.

Albumin levels were greater in the HP than in the IP formula group, which was unexpected. Previous studies with HP formulae indicated the lower protein quality of HP vs. IP formulae, which is generally associated with lower albumin serum levels and consistent with findings by Florendo et al. (2009) [43]. The unexpected difference in albumin levels in our study may be due, at least in part, to the use of different laboratories to evaluate serum albumin for the HP formula groups and IP formula groups, thus generating slightly different results. However, the serum levels for all intervention groups were well within normal reference limits [20]. In addition, no relevant differences were observed in growth indices between groups, suggesting that the observed differences in albumin levels may not be clinically important. 

The unexpected results for serum albumin levels and the additional intake of energy-containing liquids highlight one limitation of this analysis: the intervention groups compared were from two different studies and geographical regions, i.e., all data from HP formula-fed infants were from the HA study conducted in Serbia, Austria, and Germany, while IP formula-fed participants were from the BeMIM study conducted solely in Serbia. Although the statistical tests were adjusted for region, differences in cultural feeding patterns and the differences in studies may have still impacted the results. Looking at our post-hoc sample size estimation, the number of infants in one PPS group comparison (eHF vs. iPF) is probably too low to draw robust conclusions. A further limitation of the study is that more than one component differed between HP and IP. Thereby, besides the protein source, the macro- and micronutrient composition, as well as the content of long-chain polyunsaturated fatty acids, differed slightly between formulae.

The strength of this analysis lies in the similarity of the study designs: both studies were randomized, controlled studies; data were collected at almost identical timepoints; anthropometric measurements and assessments for stool characteristics were done similarly; and diets were isocaloric and comparable for protein content on the respective protein levels (standard vs. low protein). The synbiotics that were included only in the low-protein HP formula were shown to not impact infant growth [31,32] and thus are unlikely to have biased the results. Another key strength of our analysis is the use of individual-level data, allowing the harmonization of covariates, definitions, and analytical approaches.

## 5. Conclusions

The analysis demonstrated that infant formulae manufactured from extensively hydrolyzed whey protein meet infant requirements for adequate growth with similar gains in weight and z-scores, compared to infant formulae manufactured from intact protein and a reference group of breastfed infants. Based on these results, it can be concluded that infant formulae manufactured from extensively hydrolyzed whey protein are suitable and safe for infants during the first 4 months of life. Local practices in some countries, providing small amounts of energy-containing liquids (up to 50 mL) during the first 4 months of life, do not impact infant growth.

## Figures and Tables

**Figure 1 nutrients-16-00245-f001:**
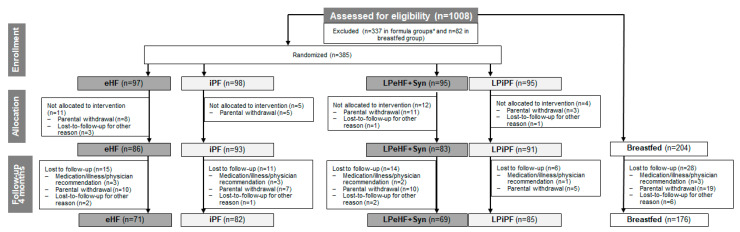
Participant disposition, randomization, and follow-up for infant formula groups compared in this analysis, and for the BF group. eHF = infant formula manufactured from extensively hydrolyzed whey protein, iPF = infant formula manufactured from intact protein, Syn = synbiotics, LP = low protein, BF = breastfeeding, * n = 210 infants received formulae from a different protein source.

**Figure 2 nutrients-16-00245-f002:**
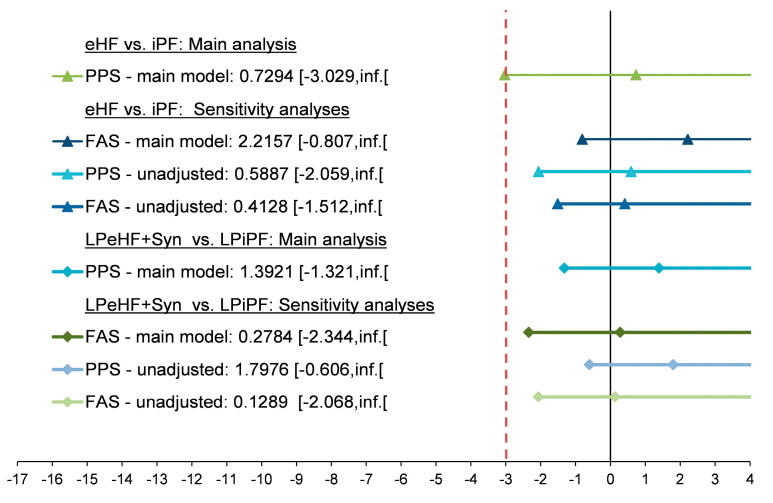
Weight gain/day differences [g/day] between infant formula manufactured from extensively hydrolyzed whey protein versus intact protein for PPS and FAS. Least square means and one-sided 97.5% confidence intervals are depicted. Main model = ANCOVA adjusted for sex, region, and baseline value. Unadjusted = ANCOVA with only the intervention group as fixed factor. The dotted line resembles the non-inferiority margin.

**Figure 3 nutrients-16-00245-f003:**
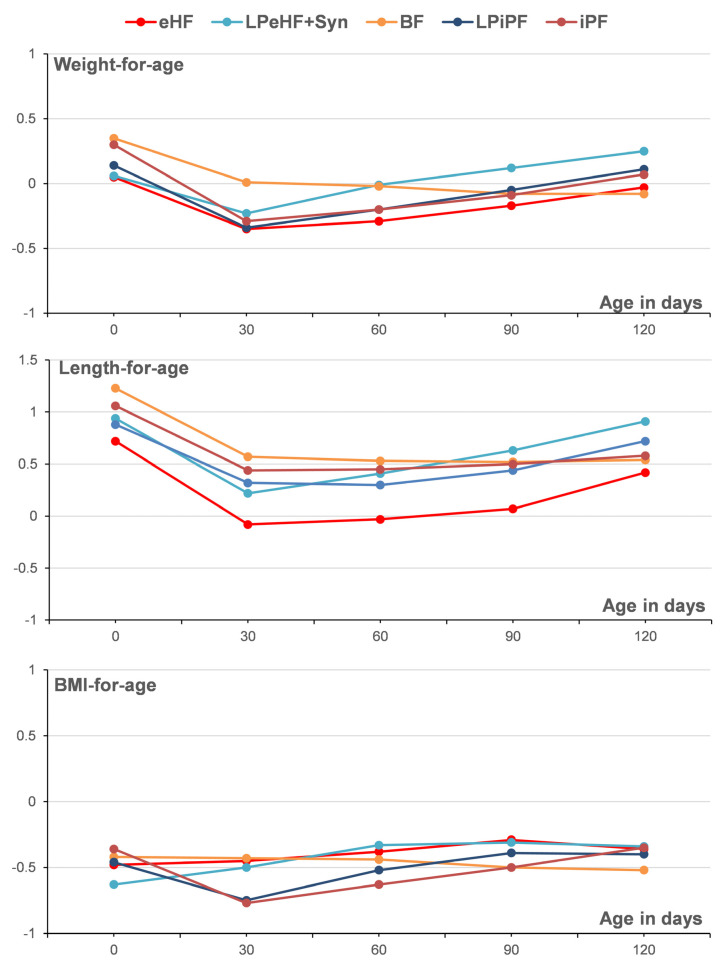
Anthropometric measurements (weight-for-age, length-for-age, and BMI-for-age) expressed as z-scores (growth standards of the WHO) (PPS). z-scores within −1 to 1 indicate age-appropriate development. BMI = body mass index, WHO = World Health Organization.

**Figure 4 nutrients-16-00245-f004:**
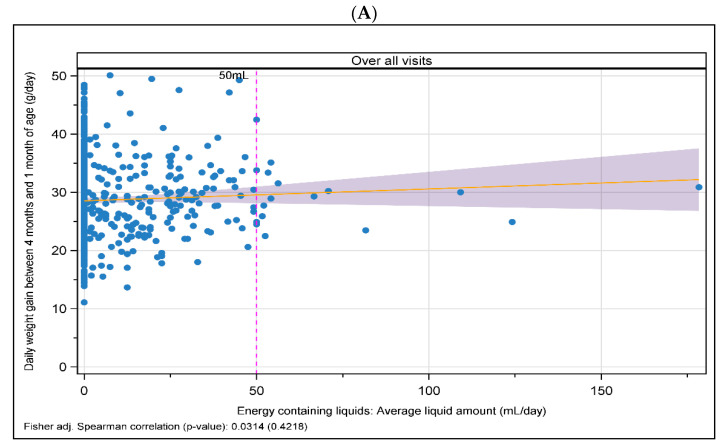
Scatterplots to correlate liquid intake and growth (FAS). (**A**) Impact of energy-containing liquid intake on weight gain between 1 and 4 months of life. (**B**) Impact of energy-containing liquid intake on weight-for-age z-scores at 1, 2, 3, and 4 months of life. Dotted line shows 50 mL cut-off.

**Table 1 nutrients-16-00245-t001:** Blood albumin and urea nitrogen at month 4. Reference ranges based on Oster 2007 [20].

	Standard Protein Group				Low Protein Group					
FAS	eHF		iPF		LPeHF + Syn		LPiPF		BF	
**Albumin**										
n	65		78		59		83		167	
Median (Q1, Q3) [g/dL]	4.1 *	(3.88,4.31)	3.7 *	(3.60, 3.90)	4.2 *	(4.04, 4.34)	3.8 *	(3.60, 3.90)	3.9	(3.70, 4.10)
n (%) of infants with values										
Within reference range	65	(100.0)	78	(100.0)	59	(100.0)	83	(100.0)	167	(100.0)
Above reference range	-		-		-		-		-	
Below reference range	-		-		-		-		-	
**Blood urea nitrogen**										
N	65		78		58		83		166	
Median (Q1, Q3) [mmol/L]	3.7	(3.16, 4.16)	2.8	(2.40, 3.10)	2.6	(2.16, 3.00)	2.5	(2.10, 3.00)	2.0	(1.50, 2.33)
n (%) of infants with values										
Within reference range	62	(95.4)	69	(88.5)	46	(79.3)	73	(88.0)	80	(48.2)
Above reference range	-		-		-		2	(2.4)	-	
Below reference range	3	(4.6)	9	(11.5)	12	(20.7)	8	(9.6)	86	(51.8)

* *p* < 0.0001 (two-sided, van Elteren test adjusted for region) for difference eHF vs. iPF and for difference LPeHF + Syn vs. LPIPF.

## Data Availability

The data presented here are available upon request from the corresponding author due to intellectual property limitations.

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
