# Peer review of "Safety and Suitability of Infant Formula Manufactured from Extensively Hydrolyzed Whey Protein Compared to Intact Protein: A Combined Analysis of Two Randomized Controlled Studies"

_nutrients, 2024, doi:10.3390/nu16020245_

Round 1

Reviewer 1 Report

Comments and Suggestions for Authors

Dear authors, 

I found the study titled 'Safety and appropriateness of infant formula made from extensively hydrolyzed whey protein compared to intact protein: a combined analysis of two randomized controlled experiments' to be very fascinating. I believe this could be of significance to the healthcare sector. 

In my opinion, it is important to address two minor points that require amendment.

1. It would be intriguing to incorporate a table that presents a side-by-side analysis of the characteristics of the two formulas (HP and IP) as outlined in Chapter 2.2 Diet. The insertion can be made either within the body of the text or within the attachment containing supplementary materials. I believe that readers would find it beneficial to have a more comprehensible and accessible explanation of the advantages of the analyzed types.

2. There are a few grammatical and writing errors that should be corrected with a re-reading of the entire content. Furthermore, kindly modify the title of the manuscript to ensure that every word is capitalized - 'Safety and Appropriateness of Infant Formula Made from Extensively Hydrolyzed Whey Protein Compared to Intact Protein: a Combined Analysis of Two Randomized Controlled Experiments'.

Comments on the Quality of English Language

There are a few grammatical and writing errors that should be corrected with a re-reading of the entire content. Furthermore, kindly modify the title of the manuscript to ensure that every word is capitalized - 'Safety and Appropriateness of Infant Formula Made from Extensively Hydrolyzed Whey Protein Compared to Intact Protein: A Combined Analysis of Two Randomized Controlled Experiments'.

Author Response

Dear Reviewer 1,

Thank you very much for your great contribution and valuable input on our manuscript.  We have integrated our point-by-point replay into your text.

Dear authors, 

I found the study titled 'Safety and appropriateness of infant formula made from extensively hydrolyzed whey protein compared to intact protein: a combined analysis of two randomized controlled experiments' to be very fascinating. I believe this could be of significance to the healthcare sector. 

In my opinion, it is important to address two minor points that require amendment.

  1. It would be intriguing to incorporate a table that presents a side-by-side analysis of the characteristics of the two formulas (HP and IP) as outlined in Chapter 2.2 Diet. The insertion can be made either within the body of the text or within the attachment containing supplementary materials. I believe that readers would find it beneficial to have a more comprehensible and accessible explanation of the advantages of the analyzed types.

Authors’answer:

A side-by-side table has been included in the supplementary materials as supplementary table 1. All other supplementary table numberings have been corrected accordingly as there is now a new supplementary table.

  1. There are a few grammatical and writing errors that should be corrected with a re-reading of the entire content. Furthermore, kindly modify the title of the manuscript to ensure that every word is capitalized - 'Safety and Appropriateness of Infant Formula Made from Extensively Hydrolyzed Whey Protein Compared to Intact Protein: a Combined Analysis of Two Randomized Controlled Experiments'.

Authors’ reply:

The title has been changed according to your suggestion. Grammatical and writing errors have been revisited and corrected.

Comments on the Quality of English Language

There are a few grammatical and writing errors that should be corrected with a re-reading of the entire content. Furthermore, kindly modify the title of the manuscript to ensure that every word is capitalized - 'Safety and Appropriateness of Infant Formula Made from Extensively Hydrolyzed Whey Protein Compared to Intact Protein: A Combined Analysis of Two Randomized Controlled Experiments'.

Authors’ reply:

The title has been changed according to your suggestion. Grammatical and writing errors have been revisited and corrected.

We wish you all the best for 2024.

Kind regards,

M. Fleddermann on behalf of all authors.

Reviewer 2 Report

Comments and Suggestions for Authors

The paper does not have any input on the problem of infant formula manufactured with hydrolyzed whey protein. The conclusion is obvious. Metabolism of infants is fast and it would be very difficult to see any differences. Besides, work is based on other researches and itself has a small impact on the issue.

Author Response

Dear Reviewer 2,

Thank you very much for your great contribution and valuable input on our manuscript.  We have integrated our point-by-point replay into your text.

The paper does not have any input on the problem of infant formula manufactured with hydrolyzed whey protein. The conclusion is obvious. Metabolism of infants is fast and it would be very difficult to see any differences. Besides, work is based on other researches and itself has a small impact on the issue.

Authors’ reply:

Thank you very much for the comment. Based on our results we provide supporting information on the safety of infant formula manufactured from hydrolysed proteins. We are aware, that there are current discussions about the efficacy e.g. prevention of allergy of those formula and the age at which those formula should be provided to the infants. This is a big field of research, which hopefully will be solved in the near future and provide new evidence for guidelines and routine paediatricians life.

Reviewer 3 Report

Comments and Suggestions for Authors

This study investigated the safety and suitability of infant formula manufactured from extensively hydrolyzed whey protein compared at 2 levels of protein standard and low protein formulae manufactured from intact cow’s milk with comparable protein content.

The authors successfully proved that the infant formula manufactured from extensively hydrolyzed protein meets infant requirements for adequate growth and does not raise any safety concerns.

It is meaningful work, but here are some minor questions and suggestions the authors may consider in this work.

·         Lines 50-51, it should be explained how “the level of hydrolysis, the protein source, and other components may affect the 50 safety and tolerance of different HP formulae.”

·         Infants consumed energy-containing liquids, what are these liquids? I understand that the authors said “For more information on infant formulae composition see Ahrens et al. (2018) [16] and Fleddermann et al. (2014) [17]”; however, it should be mentioned in the materials and methods section.

·         Line 115-116, “a maximum of 50 mL tea (or water) intake per day…” not clear does that mean 4-month-old babies were given 50 mL tea? Why? It should be clarified and explained.

·         I would suggest that the study population section should be included in the materials and methods not to the results.

·         The majority of results are in supplementary figures and tables, even though they are available online, it would be better to include at least some tables in the manuscript.

·         In Lines 280, 294, and 309, “data not shown” it should be at least included in the supplementary material.

·         Conclusion: the valuable information is not enough. It needs to be rewritten. 

Author Response

Dear Reviewer 3,

Thank you very much for your great contribution and valuable input on our manuscript.  We have integrated our point-by-point replay into your text.

- Lines 50-51, it should be explained how “the level of hydrolysis, the protein source, and other components may affect the 50 safety and tolerance of different HP formulae.”

Authors’ reply:

For lines 54-56 „The level of hydrolysis, the protein source, and other components may affect the safety and tolerance of different HP formulae, thus the extrapolation of results from one HP formula to another is not accepted by regulators“ […] we have clarified, that this is a regulatory statement according the EU directive 2016 on the composition of infant formula.

- Infants consumed energy-containing liquids, what are these liquids? I understand that the authors said “For more information on infant formulae composition see Ahrens et al. (2018) [16] and Fleddermann et al. (2014) [17]”; however, it should be mentioned in the materials and methods section.

Authors’ reply:

Energy containing liquids are tea with sugar or juice. The possibility for documentation of liquids have been addressed in sec. 2.2 Diet.

- Line 115-116, “a maximum of 50 mL tea (or water) intake per day…” not clear does that mean 4-month-old babies were given 50 mL tea? Why? It should be clarified and explained.

Authors’ reply:

The sentence may have been confusing and we have changed it to „In PPS, only data from participants complying with the predefined conditions, completion of the intervention period up to 4 months of life, no intake of other infant formula than study intervention and breastfeeding at a maximum of once daily, were included.”

According to local routine in study countries, some infants receive additional liquids like water or tea to the infants. Based on this we have adapted the wording on liquids “ According to local practice in study countries, some infants receive additional liquids like tea or water next to breastmilk or infant formula during the first 4 months of life, which was limited to a maximum amount of 50mL per day to be still valid for inclusion into PPS.”

- I would suggest that the study population section should be included in the materials and methods not to the results.

Authors’ reply:

The former section 3.1 is now moved to chapter 2 and is now located under section 2.5 Study Population.

- The majority of results are in supplementary figures and tables, even though they are available online, it would be better to include at least some tables in the manuscript.

Authors’ reply:

Within the main text, we had 4 Figures and 1 Table presenting the key results of the publication. This is a guidance from the journal to keep the main manuscript clear and focused. We have critically reviewed the content of the Tables and Figures in the main text and Supplementary material and present the key results in the main text. We welcome your input, which Table or Figure should be moved from the Supplementary material to the main text.

- In Lines 280, 294, and 309, “data not shown” it should be at least included in the supplementary material.

Authors’ reply:

Ad line 280/ manuscript section 3.5.1. Adverse Events: To see numbers of infants affected by overweight we introduced a further row in supplementary table S8 Adverse events (FAS). The added row with PT ‚Overweight‘ specifies that 7 adverse events in 7 infants in the HP groups have been affected by overweight, but none in the intact protein groups.

Ad Line 294/ manuscript section 3.5.2. Stool characteristics: To support the sentence that statistically significant different stool color patterns have been seen in the low protein groups (LPeHF+Syn vs LPiPF) at all timepoints in FAS we added a supplementary table S10 with descriptive statistics and p-values. The reference to PPS was deleted as we haven’t presented results for frequency and consistency for PPS either.

Line 309/ manuscript section 3.5.2. Stool characteristics: Description of stool consistency results can be depicted in supplementary table S10. Of note we also added descriptive statistics and p-values of stool frequency in this table.

  • Conclusion: the valuable information is not enough. It needs to be rewritten. 

Authors’ reply:

We have added additional explanations to the conclusions section.

We wish you all the best for 2024.

Kind regards,

M. Fleddermann on behalf of all authors.

Round 2

Reviewer 2 Report

Comments and Suggestions for Authors

The presented version can be published.